# Bilirubin Photoisomers in Neonatal Jaundice

**DOI:** 10.3390/ijms262110791

**Published:** 2025-11-06

**Authors:** Dennis Lindqvist, Magnus Hansson, Mercy Thomas, Christian V. Hulzebos, Libor Vitek, Andries Blokzijl, Miranda van Berkel

**Affiliations:** 1Department of Clinical Chemistry, Karolinska University Hospital, 171 64 Stockholm, Sweden; magnus.d.hansson@regionstockholm.se (M.H.); andries.blokzijl@regionstockholm.se (A.B.); 2Department of Laboratory Medicine, Karolinska Institutet, 171 77 Stockholm, Sweden; 3School of Health Sciences, Swinburne University of Technology, Melbourne, VIC 3122, Australia; mercythomas@swin.edu.au; 4Murdoch Children’s Research Institute, Parkville, VIC 3052, Australia; 5Division of Neonatology, Department of Pediatrics, Beatrix Children’s Hospital, University Medical Centre Groningen, 9713 GZ Groningen, The Netherlands; c.v.hulzebos@umcg.nl; 6Institute of Medical Biochemistry and Laboratory Diagnostics, and 4th Department of Internal Medicine, 1st Faculty of Medicine, Charles University and General University Hospital in Prague, 121 08 Prague, Czech Republic; 7Department of Laboratory Medicine, Radboud University medical center, 6525 GA Nijmegen, The Netherlands; miranda.vanberkel@radboudumc.nl

**Keywords:** hyperbilirubinemia, mass spectrometry, newborn infant, photoisomers, phototherapy

## Abstract

Phototherapy is the standard treatment for neonatal hyperbilirubinemia. During phototherapy, the highly lipophilic bilirubin is converted into more hydrophilic photoisomers, which can be more easily excreted from the body. This process typically lowers bilirubin levels to non-harmful concentrations. However, despite decades of research into the formation and role of bilirubin photoisomers, methodological limitations and the compound’s complex biochemistry have hindered comprehensive understanding. This review provides an updated overview of current knowledge on bilirubin photoisomers, including their basic chemistry, analytical quantification, clinical relevance, and future research directions. Improved insight into the mechanism of photoisomer formation and kinetics may inform optimization of phototherapy parameters, including light intensity and wavelength, and offer additional indicators of treatment efficacy beyond total bilirubin concentration. Advances in sensitive and standardized mass spectrometry techniques now enable more accurate measurement of different bilirubin isomers and serve as a first step towards a deeper insight into the clinical relevance of photoisomers.

## 1. Introduction

Phototherapy (PT) is universally used to treat newborn infants with neonatal hyperbilirubinemia (NHB). Since its introduction in the 1950s, PT has gradually replaced exchange transfusion as the standard treatment for NHB [1]. This was initiated by work published in the Lancet in 1958 by Cremer, Perryman, and Richards, which demonstrated that both sunlight and blue fluorescent light reduced bilirubin concentrations in newborns [1]. However, small sample sizes and a lack of appropriate controls led to some skepticism among clinicians who feared potential toxic by-products and were uncertain of the underlying biochemical processes. Ferreira and colleagues were the first to coin the term “phototherapy” when they similarly observed the clinical efficacy of light therapy in treating NHB [2]. This was followed in 1968 by a well-controlled study that proved the efficacy of PT and greatly influenced clinical practice [3]. Already the standard of care in the United States (U.S.) by the mid-1970s, a later large randomized controlled trial (NICHD PT trial, 1974–76) further reinforced the efficacy and safety of PT [4]. Thereafter, PT was recommended by the American Academy of Pediatrics for healthy term infants with NHB [5].

However, re-examination of the NICHD data raised concerns that aggressive lowering of bilirubin might increase mortality in extreme low birth weight (ELBW) infants. This led to a large multicenter randomized control trial being conducted to clarify the balance between the neurodevelopmental benefits and mortality risks associated with different PT strategies [6]. While aggressive PT (initiating treatment at lower thresholds) reduced the risk of long-term neurodevelopmental impairment in ELBW infants, it appeared to increase the risk in the most fragile ELBW subgroups, those with birthweight ≤ 750 g, who required mechanical ventilation. This indicates that there might be a trade-off for better neurodevelopmental outcomes for survivors, but possibly at the cost of increased mortality. These findings highlight the critical need for individualized treatment approaches and the need to tailor new treatment modalities carefully to specific patient subgroups [6].

The mechanisms underlying PT involve the exposure to photons of bilirubin circulating primarily bound to albumin in the intravascular compartment and also present in the skin. In the skin, where bilirubin is directly accessible to photons, it undergoes structural and configurational changes that lead to the formation of photoisomers and bilirubin photooxidation (BOX) products including tripyrrolic biopyrrines, dipyrrolic propentdyopents, and monopyrrolic BOXes [7]. The formation of BOX products has been extensively reviewed elsewhere [8], and is beyond the scope of this review. While some photoisomers can revert to bilirubin, others, such as lumirubin (LR), are practically irreversible, water soluble, and rapidly excreted [9,10].

The efficacy of PT is traditionally monitored by measuring total serum bilirubin (TSB). However, TSB reflects the overall bilirubin pool, including a fraction of its photoisomers, and does not provide insight into the specific molecular changes induced by light exposure [11]. This limits its ability to capture the complete biochemical effects of PT. A more detailed understanding of the kinetics of photoisomer formation, particularly LR, and of eventual neurotoxicity of photoisomers could help refine PT strategies. Measuring LR, for instance in urine, may serve as a non-invasive marker of treatment efficacy. This could be especially valuable in preterm infants, for whom minimizing blood sampling is important [12]. Understanding the kinetics of isomer formation and their relationship with TSB concentrations may help optimize PT by refining wavelength, irradiance, and treatment duration.

This review summarizes the major molecular changes that bilirubin undergoes during PT in newborns with NHB. It also discusses established and emerging methods for quantifying photoisomers and explores how these markers may contribute to the improved monitoring and optimization of PT in newborn infants with NHB.

## 2. Chemistry of Photoisomers

Under light exposure, unconjugated bilirubin (UCB; the native bilirubin without any added glucuronide moieties) undergoes two main photochemical transformations: configurational and constitutional isomerization. These processes enable the clearance of UCB via urine and bile, with the rate of excretion depending on the route and the reversibility of the process. To clarify the biochemical pathways and the formation of the resulting photoproducts, the nomenclature of these different molecules is explained in detail. Table 1 summarizes the classification of photoisomers.

Bilirubin, more precisely referred to as 4*Z*, 15*Z*-bilirubin IXα (hereafter referred to as *ZZ*-BR, *zusammen: Z*), is formed during the catabolism of heme B. This process involves heme oxygenase-catalyzed cleavage of the α-methine bridge in protoporphyrin (IX; PPIX), followed by reduction in the resulting biliverdin by biliverdin reductase. *ZZ*-BR is a tetrapyrrole molecule with a molecular weight of 584.7 g/mol. Each of its two inner pyrrole units carries a propionic acid substituent that can form intramolecular hydrogen bonds with the NH groups and C=O groups of the opposing dipyrrinone moiety. Through this interaction, *ZZ*-BR adopts a V-shaped (ridge-tile) structure with an angle of approximately 100° (see Figure 1). The formation of these intramolecular hydrogen bonds renders *ZZ*-BR poorly water-soluble, making excretion difficult unless conjugated. When unbound to albumin, its high lipophilicity facilitates passage through the blood–brain barrier (BBB) [13,14]. Under normal conditions, *ZZ*-BR exists in dynamic equilibrium between two distinct conformational isomers.

Conformational isomers share the same constitution and spatial arrangement of covalent bonds but differ in their three-dimensional orientation in space. Due to the inherent asymmetry of the *ZZ*-BR structure, two enantiomeric (i.e., mirror image) conformational isomers are formed when the ridge-tile structure is adopted (see Figure 1). In achiral solutions, these enantiomers exist as a racemic mixture. However, binding to the chiral albumin molecule is enantioselective, resulting in preferential binding of one enantiomer (*P*) over the other (*M*) (see Figure 1) [15].

*Configurational isomers* share the same constitution and arrangement of atoms in the structure but differ in the spatial configuration. During PT, *ZZ*-BR can rearrange into *ZE*-BR (*zusammen, entgegen*), *EZ*-BR, and *EE*-BR. These isomers are a type of diastereomers because they are not related as mirror images. In the bilirubin literature, they are sometimes referred to as “geometric isomers”; however, this is considered an obsolete term according to the International Union of Pure and Applied Chemistry (IUPAC) [16]. The rotation of the terminal pyrrole rings during photoisomerization causes the heteroatoms to be oriented away (*entgegen*; *E*) from the central pyrroles (see Figure 1). This arrangement disrupts intramolecular hydrogen bonding with the propionic acid groups. As a result, the acid groups as well as the heteroatoms of the terminal pyrroles in the *E* configuration are exposed for interactions (e.g., hydrogen bonding) with the surrounding medium (water). This increased accessibility makes these isomers more water-soluble, and thus more easily excreted.

*Constitutional isomers* share the same molecular weight and chemical formula but differ in the arrangement of the atoms within the structure. Although synonymous with “structural isomer”, the latter term has been used somewhat arbitrarily in the bilirubin literature, whereas “constitutional isomer” is the designation recommended by the IUPAC [16]. During PT, the constitutional isomer LR, also known as cyclobilirubin, is formed. LR is believed to be the most readily excreted isomer formed during PT. It occurs in two different configurational forms, *Z*-LR and *E*-LR, each possessing stereogenic centers that give rise to enantiomeric pairs (see Figure 1). Other constitutional isomers, such as BR IIIα and BR XIIIα, are not products of heme catabolism but can arise through the conversion of BR IXα. This conversion is hampered by the presence of albumin [17] resulting in low in vivo levels of BR IIIα and BR XIIIα with questionable clinical relevance. Slightly higher yet still low levels have been reported in commercial reference materials [18], which may introduce analytical bias in end-user assays.

**Table 1 ijms-26-10791-t001:** Classification and properties of bilirubin and its isomers.

Isomerization	Product	Reversibility	Solubility	Excretion Route	Clearence * /Half-Life	Clinical Role	Ref
Conformational	*P* and *M* *ZZ*-BR	Dynamic Equilibrium	Low	Bile, following conjugation	2% day^−1^ preterm 11% day^−1^ full-term	Slow excretion Passage through BBB	[14,15,19]
Configurational	*ZE*-BR *EZ*-BR (*EE*-BR)	Reversible	Moderate	Urine + bile	t_1/2_ = 15 h	Enhances excretion	[20,21,22]
Constitutional	*Z*-LR (*E*-LR)	Practically Irreversible	Moderate High	Urine + bile	t_1/2_ = 2 h	Main route for permanent clearance	[9,10,12,22]

* This represents a calculated clearance that includes both formation and excretion, following the 4th–5th postnatal day in preterm infants and the 3rd–4th postnatal day in full-term infants, respectively.

## 3. Clinical Implications of PT on Photoisomers

### 3.1. Wavelength Considerations in PT

Current clinical guidelines recommend the use of intensive blue light within the 460–475 nm spectrum, aligned with the absorption maximum of bilirubin [23,24]. Interestingly, Ebbesen and colleagues reported that the optimal efficacy of PT for neonatal hyperbilirubinemia occurs at a slightly higher wavelength, approximately 478 nm, resulting in a 55.6% reduction in ZZ-BR compared to 44.2% at 459 nm [25]. This improvement was attributed to more efficient LR formation, reduced competitive absorbance by hemoglobin and melanin, and decreased light backscattering at longer wavelengths. However, the increased LR formation observed at longer wavelengths is accompanied by a reduced formation of *ZE*-BR [25], highlighting that the composition of the photoisomer generated during PT depends on the wavelength of the light source used (Figure 2 summarizes the reactions and products formed during PT). This dependency can be explained by the bichromophoric nature of BR.

BR is not aromatic through the central methyl bridge; it contains two separated chromophores, and two light-absorbing dipyrrole units (see Figure 1). These two subunits differ only structurally in the position of the methyl and vinyl substituents of the terminal pyrroles [27]. However, this subtle difference is sufficient to favor excitation of one chromophore over the other depending on the wavelength. Although excitation energy can be transferred between the chromophores, relaxation and isomerization occur rapidly enough that the composition of photoproducts remains wavelength-dependent [27]. The *ZE*-BR isomer is preferentially formed at shorter wavelengths, whereas the *EZ*-BR isomer, from which *Z*-LR is formed, is favored at longer wavelengths. Indeed, higher production rates of *EZ*-BR were reported in human neonates exposed to turquoise light (497 nm) compared with blue light (459 nm) [28]. Green light within the 500–520 nm range appears particularly effective in generating LR [12]. Different wavelengths used in PT and their relation to the absorption spectrum of bilirubin is displayed in Figure 3.

Formation of *Z*-LR from *EZ*-BR proceeds more rapidly than the formation of *EZ*-BR from *ZZ*-BR [21]. Therefore, the rate-limiting step in LR generation does not appear to be the photocyclization itself, but rather the configurational isomerization of native BR to the *EZ*-isomer. The cyclization reaction forming LR is, however, thought to proceed more slowly than the configurational isomerization to *ZE*-BR in vivo [9], and it has a lower quantum yield, Φ_ZZ-ZE_ ≈ 0.2, compared to Φ < 0.005 for lumirubins (for BR/HSA, 265 nm, 22 °C) [29,30]. The relative quantum yield also varies with wavelength [27]. Despite its slower formation rate, the excretion of LR in vivo is faster, with a half-life of approximately 2 h [9] compared to 15 h for *ZE*-BR in serum [20], and thus accounts for the greater overall efficacy of longer wavelength PT [25] (Table 1). In conclusion, although LR is formed at a slower rate in vivo, its rapid excretion still makes LR the most important photoisomer in the elimination of UCB.

**Figure 3 ijms-26-10791-f003:**
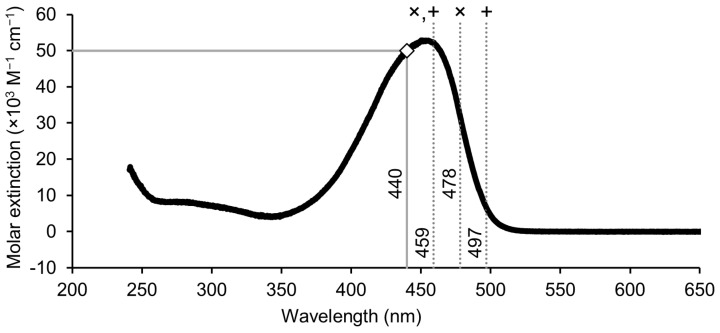
Absorption spectrum of unconjugated BR in chloroform [31,32]. Dashed gray lines indicate the wavelength used for PT in various studies (×: ref [25], +: ref [28]). The solid gray line with a white square displays the measured extinction coefficient at 440 nm in phosphate buffer pH 7.4 [33]. The spectrum was scaled to match the extinction coefficient of 50,000 M^−1^ cm^−1^ at 440 nm.

### 3.2. Photoisomer–Albumin Interaction

Bilirubin in the circulation is predominantly bound reversibly to albumin. Human serum albumin (HSA) binds *ZZ*-BR with high affinity [34], where *ZZ*-BR is thought to primarily interact with domain IIA. In contrast, *ZE*-BR binds weakly to domain IB of human albumin, aiding excretion of its more water-soluble form. Crystallography at 2.42 Å shows it binds in an L-shaped pocket within subdomain IB [35]. Beyond its direct binding effects, HSA also influences the formation of bilirubin isomers. The binding behavior of bilirubin to HSA differs compared to that of other species due to structural variations in albumin’s binding site. The rate constant of *Z*-LR formation from *EZ*-BR is significantly higher in the presence of albumin from humans and rhesus monkeys than with albumin from other species including bovine, rat, chicken, dog, and rabbit [36].

The role of albumin in inhibiting neurotoxicity has been demonstrated in various animal models. Infusion of HSA increases the efficacy of PT in reducing UCB concentrations in both chronic Gunn rat models of Crigler–Najjar disease in which the bilirubin-conjugating enzyme UGT1A1 is absent, and in acute hemolytic models of Gunn rats. In the acute neonatal jaundice rat model, HSA infusion combined with PT eliminated intrathecal bilirubin precipitation, whereas PT alone failed to do so [37]. By crossing Gunn rats with Nagase analbuminemic rats [38], an albumin deficient and jaundiced strain of rats (AJR) was established [13]. These AJR rats displayed higher brain bilirubin content and died of kernicterus within three weeks despite lower serum bilirubin concentrations than parental Gunn rats, which typically die at 1–2 weeks after birth [13]. Consistent with these findings, in a mouse model of severe neonatal hyperbilirubinemia, daily infusions of human albumin prevented neurological impairment and mortality [39] by increasing plasma bilirubin binding capacity, mobilizing bilirubin from tissues to plasma, and reducing the bilirubin deposition in the forebrain and cerebellum [39].

Although albumin infusions show consistent benefits in animal models, randomized trials in humans have generated mixed results. Some studies reported reduction in UCB concentrations and shorter PT duration when primed with albumin pre-exchange transfusion [40,41], whereas others did not identify a statistically significant effect on post-exchange TSB concentration or on the number of exchange transfusions [42,43]. In preterm infants, the combination of low albumin concentrations and an immature blood–brain barrier increases the risk of kernicterus [44]. A serum albumin concentration below 3.0 g/dL is considered a risk factor for neurotoxicity that lowers the phototherapy threshold according to AAP guidelines [45]. As such, the AAP recommends quantifying albumin concentration as part of the escalation of care process and considering bilirubin-to-albumin ratios when determining the need for exchange transfusion. Although albumin assessment provides important guidance in evaluating the need for PT or ET, albumin infusions themselves are not routinely recommended as part of clinical escalation [45].

### 3.3. Toxicity of Photoisomers

While PT effectively lowers TSB in newborns, questions persist regarding the toxicity of the resulting photoisomers and other photooxidation products [46]. In vitro studies generally indicate that these isomers, including *ZE*-BR, *EZ*-BR, and *Z*-LR, are less toxic than native *ZZ*-BR [47]. They exhibit minimal effects on cell viability, induce fewer mitochondrial and oxidative stress changes, and *ZE*-BR may even offer neuroprotective benefits [48]. However, LR was demonstrated to induce neuroinflammation under experimental conditions [7].

Animal studies have raised additional concerns regarding potential adverse effects of excessive or prolonged light exposure, including phototoxicity and tissue damage [7]. Findings from a UGT1A1-deficient mouse model demonstrated that while PT reduces TSB concentrations, it may also be associated with neurological alterations, possibly due to the accumulation or effects of certain isomers. LR in particular has been shown to influence early neural development by affecting neural stem cell morphology and protein expression, suggesting that individual isomers may have distinct safety profiles [49].

Accurately assessing the true toxicity of these photoisomers is challenging due to methodological limitations [48]. Inconsistent experimental conditions, such as variations in light sources, bilirubin concentrations, and cell types, contribute to conflicting results. Moreover, the absence of reliable in vivo models limits definitive assessment of relative toxicity and blood–brain barrier permeability. These challenges underscore the urgent need for standardized and robust research methodologies.

## 4. Quantification of Bilirubin and Its Photoisomers

### 4.1. Impact on Routine Clinical TBIL Assays

Routine methods for measuring TSB, often referred to as total bilirubin assays (TBIL), include several analytical approaches. These include direct spectrophotometric methods which measure bilirubin absorbance at specific wavelengths; oxidation methods using vanadate or bilirubin oxidase, which determine the decrease in bilirubin absorbance following its oxidation to biliverdin; and colorimetric (diazo) methods where bilirubin reacts with diazotized sulfanilic acid or similar reagants to form azobilirubin. The so-called direct bilirubin assays (DBIL) quantify bilirubin forms that react without the need for accelerators. Typically, these include mono- and diconjugated forms (i.e., BR with one or two glucuronic acid moieties attached to the propionic acid subunits). The δ-bilirubin (BR covalently bound to albumin) generally mimics the reactivity of the conjugated forms. The degree to which δ-bilirubin contributes to DBIL values varies with assay method, reagent composition, pH, buffer, and reaction time [50,51]. The concentration difference obtained between TBIL and DBIL assays is used to quantify the unconjugated form of bilirubin.

Photoisomers of *ZZ*-BR may influence the measurement of TBIL and DBIL, because these photoproducts maintain key chemical features, such as exposed pyrrole rings and reactive functional groups, which enable them to maintain reactivity in TBIL and/or DBIL assays. In addition to UCB and mono- and diconjugated forms of BR, both the configurational isomers and constitutional isomers (i.e., LR) can act as substrates in the vanadate method [52,53], and the bilirubin oxidase method [54,55,56], whereas the configurational isomers but not LR can act as substrates in the diazo method [11,22]. Current clinical methods for the measurement of TBIL cannot quantify bilirubin photoproducts, and it remains unclear which routine analytical tools could be implemented in clinical practice to quantify these compounds.

### 4.2. Quantitative Analysis of Photoisomers

Several (high pressure) liquid chromatography, (HP)LC, and LC–MS methods have been developed to separate and detect *ZZ*-BR and its photoisomers [57,58,59,60]. However, the inherent instability of the BR photoisomers means that there is a lack of commercially available standards for quantitative analysis. Quantitative data in the literature are generally based on absorption and assumptions on the molar extinction coefficient for the various isomers. Itoh and colleagues derived relative molar extinction coefficients for *ZE*-BR (0.81), *EZ*-BR (0.54), *Z*-LR (0.46), and *E*-LR (0.39) relative to that of *ZZ*-BR (1.0) at 455 nm in the mobile phase buffer (acetonitrile, phosphate buffer, dimethylformamide). These were determined by stepwise degradation of *ZZ*-BR, assuming that the sum of all detected isomers would be equal to the starting concentration of *ZZ*-BR [57].

Procedures have also been developed for isolating and purifying *Z*-LR formed via photolysis of *ZZ*-BR, receiving a molar extinction coefficient of approximately 33,000 M^−1^ cm^−1^ in HSA solution at 453 nm [10]. However, even though thin layer chromatography is used to separate the products and isolate *Z*-LR, the final product is not entirely pure [58]. The extinction coefficient can be compared to that of *ZZ*-BR in the phosphate buffer, ca. 50,000 M^−1^ cm^−1^ at 440 nm [33,61], which supports the previously derived relative coefficients [57], suggesting that they are likely slightly lower.

While using these molar extinction coefficients likely generates adequate data with the right order of magnitude, full quantitative analytical methods to measure photoisomers of *ZZ*-BR are still lacking. Even for LC–MS, it may remain necessary to quantify their calibrator with spectrophotometric assays using these extinction coefficients to achieve quantitative data [58].

## 5. Future Perspective

### Analysis of Urinary Photoisomers

To date, TSB remains the sole clinical biomarker used to assess PT efficacy. However, exclusive reliance on TSB requires repeated blood sampling, which increases the risk of iatrogenic anemia among neonates in intensive care units. This limitation has generated growing interest in additional biomarkers, such as urinary LR. Unlike configurational isomers that can revert to native *Z,Z*-bilirubin, LR is structurally modified and irreversibly excreted via bile and urine [12]. As LR formation directly reflects active bilirubin photolysis and clearance, urinary LR measurement may present a valuable adjunct indicator of PT efficacy.

Urinary LR is attractive as a complementary biomarker due to the non-invasive nature of urine sampling, especially for preterm and low birth weight neonates who are highly vulnerable to the adverse effects of repeated blood sampling [62]. Furthermore, recent analytical advances including LC–MS and oxidation-based fluorescence assays have emerged to quantify urinary LR [58,63]. Urinary LR quantification might be valuable to optimize PT parameters such as wavelength, irradiance intensity, and duration, supporting a more individualized therapeutic approach [12]. Despite these advantages, several challenges remain. These include variability in urine concentration and the timing of urine collection. Urinary excretion of photoisomers may also lag or occur asynchronously with changes in TSB concentrations.

Future studies should therefore focus on validating LC–MS-based urinary LR assays in newborns, establishing normative reference ranges and evaluating their utility in real-time monitoring of PT efficacy. Such work could inform clinical decision-making regarding treatment escalation or the discontinuation of PT. Hence, we recommend further investigation of the kinetics of bilirubin and its photoproducts during PT, with an emphasis on applying modern methodologies and achieving robust clinical validation.

## Figures and Tables

**Figure 1 ijms-26-10791-f001:**
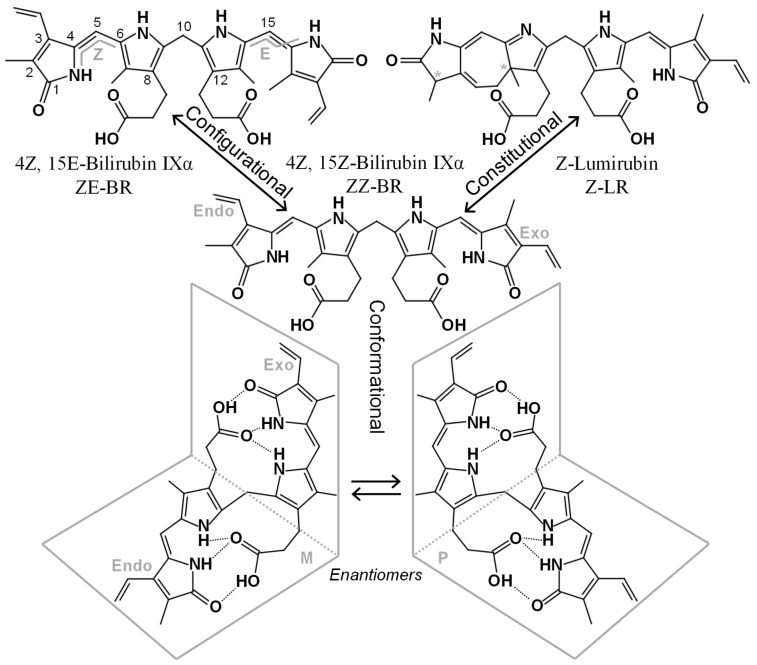
Structures and nomenclatures of some BR isomers. ZZ-BR (center), its asymmetry is highlighted with the endo-vinyl dipyrrinone to the left, and the exo-vinyl dipyrrinone to the right. Below, two enantiomeric conformers of *ZZ*-BR are depicted, formed through intramolecular hydrogen bonding. The top left depicts the configurational isomer ZE-BR with numbering and the *E*, *Z* configuration indicated. The top right depicts the constitutional isomer Z-LR, which contains two stereogenic centers (*), each producing an enantiomeric pair.

**Figure 2 ijms-26-10791-f002:**
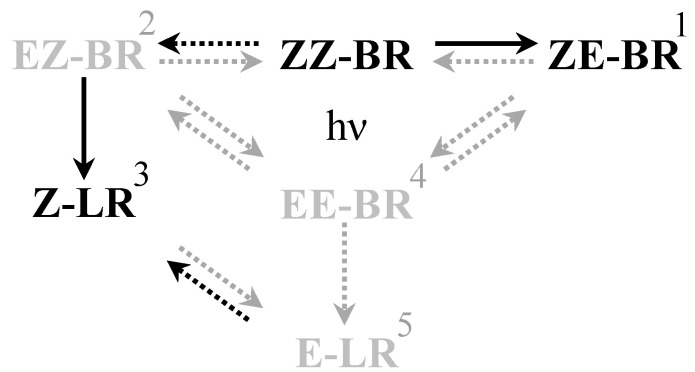
Schematic overview of the isomerization reactions occurring during PT. Solid arrows are of greater importance than dashed arrows. Black arrows and isomers are of higher importance than gray. (1) *ZE*-BR is the major photoisomer found in blood following PT [20]. (2) *EZ*-BR is of less importance due to slower formation rate and the practically irreversible nature of the conversion to *Z*-LR [10], making it more of an intermediate [21]. (3) *Z*-LR is more important due to its rapid excretion which makes it a major source of bilirubin reduction during PT [9]. (4) *EE*-BR is the thermodynamically least favorable of the configurational isomers [26], hence shifting the equilibrium largely towards *EZ*-BR and *ZE*-BR. (5) E-LR is less favorable for formation than *Z*-LR.

## Data Availability

No new data were created or analyzed in this study.

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
