# Peer review of "Bilirubin Photoisomers in Neonatal Jaundice"

_ijms, 2025, doi:10.3390/ijms262110791_

Round 1

Reviewer 1 Report

Comments and Suggestions for Authors

The manuscript addresses an important biomedical topic: monitoring the effectiveness of phototherapy of neonatal jaundice. It may be published in the journal after resolving the following items:

  1. The expressions "unconjugated” and “conjugated” bilirubin need to be explained to the inexperienced reader.
  2. For illustration, the manuscript should include the absorption spectra of bilirubin photoisomers (including lumirubin), with indication of the wavelengths used in phototherapy. At least some quantum yields of photoisomerization processes should also be provided.
  3. Along with bilirubin photoisomers and lumirubin, bilirubin photooxidation products (that can be detected in urine) can also serve as biomarkers of phototherapy effectiveness. More comprehensive data on the effectiveness of their formation during phototherapy is required.

Author Response

Comment 1. The expressions "unconjugated” and “conjugated” bilirubin need to be explained to the inexperienced reader.

Response1:We agree with the reviewer and have added text to clarify, see lines 85 and 271. 

Comment 2:For illustration, the manuscript should include the absorption spectra of bilirubin photoisomers (including lumirubin), with indication of the wavelengths used in phototherapy. At least some quantum yields of photoisomerization processes should also be provided. 

Response 2: Thank you for the thoughtful observation. We have added a figure with the absorption spectra for bilirubin where we have marked wavelength used in the different studies reviewed in the discussion (Figure 3). We have also added some quantum yield data within the discussion regarding formation of photoisomers, see lines 188-189. 

Comment 3: Along with bilirubin photoisomers and lumirubin, bilirubin photooxidation products (that can be detected in urine) can also serve as biomarkers of phototherapy effectiveness. More comprehensive data on the effectiveness of their formation during phototherapy is required.

Response 3:  

Thank you for this constructive feedback. Although the formation of BOX products has been extensively reviewed elsewhere, their clinical relevance, kinetics in a body and possible biological activities are largely unknown, as also the mechanisms of their formation during phototherapy of neonatal jaundice. Therefore, we only included a brief information on these photooxidation products and chose to restrict the review to only the photoisomers as this is comprehensive and valuable in itself. On Line 66, we have included a statement regarding this feedback. “The formation of BOX products has been extensively reviewed elsewhere [8], and is beyond scope of this review.”

Reviewer 2 Report

Comments and Suggestions for Authors

In this manuscript, the authors have presented “Bilirubin Photoisomers in Neonatal Jaundice”. The results are interesting, but needs to be explained briefly and logically.

Comment 1:

The manuscript (Page 6, lines 173-175) claims the two light-absorbing dipyrrole subunits of bilirubin "only differ structurally in the position of the methyl and vinyl substituents". This is a severe oversimplification and chemically inaccurate. The two chromophores are fundamentally non-equivalent due to the IXα structure (different methyl, vinyl, and propionic acid groups on C8,C12,C13,C17), which is the critical reason for the wavelength-dependent photochemistry discussed immediately afterward.

Comment 2:

The manuscript (Page 9, lines 313-314) states that lumirubin (LR) is "irreversibly excreted via bile and predominantly via urine". While LR is water-soluble and cleared via the kidney, the primary physiological route for LR clearance in infants with normal liver function is via bile. The claim of "predominantly via urine" is generally considered inaccurate in the clinical context.

Comment 3:

The manuscript (Page 8, lines 272-274) includes δ-bilirubin (bilirubin covalently bound to albumin) in the list of species that react in the Direct Bilirubin (DBIL) fraction. This is conceptually confusing. δ-bilirubin does not react directly in the traditional diazo method (which defines DBIL) due to its strong covalent binding to albumin. It is measured in the "Direct" fraction only because its clearance pathway mimics conjugated bilirubin, but describing it as a reactive form in routine assays is chemically imprecise and misleading.

Comment 4:

The manuscript (Page 8, line 279) states that in diazo methods, "only the configurational isomers" act as substrates. This is incorrect. Conjugated bilirubin (mono- and di-glucuronides) is the primary target that reacts in the direct fraction of the diazo method. While configurational isomers interfere with this measurement, they are not the only substrates, which fundamentally misrepresents the chemistry of the standard clinical assay.

Comment 5:

The authors suggest that the unique structure of human albumin, which favors Z-LR formation, "has led some authors to suggest the possibility of an evolutionary step" to reduce the effects of neonatal hyperbilirubinemia. This is a highly speculative, teleological interpretation of natural variation and oversimplifies evolutionary processes, presenting an assumption as a widely accepted "suggestion" based on minor protein differences (Page 6, lines 207-210).

Comment 6:

The manuscript (Page 8, lines 290-291) claims the "inherent instability" of photoisomers means there is a "lack of available standards" for quantitative analysis. While purifying and maintaining pure, commercial standards for all isomers is challenging, this statement is too strong. Purified Z-LR standards are generated and used as critical reference points in the field, making the assertion of a total "lack" a misrepresentation of the practical state of the art.

Comment 7:

The manuscript (Page 6, lines 194-196) claims that LR binding to HSA's subdomain IB "results in the increased binding capacity of albumin to UCB". The clearance of LR and ZE-BR frees up the secondary binding site (IB), which is beneficial for neuroprotection, but this process does not increase the total molar capacity or the affinity of the primary site (IIA) for unconjugated bilirubin (UCB). The claim overstates the observed neuroprotective effect.

Comment 8:

The manuscript (Page 6, lines 169-170) explains the equilibrium shift by stating that EE-BR is the "thermodynamically least favorable" configurational isomer. While EE-BR is the least stable (having no intramolecular hydrogen bonds), the composition of the PSS is governed by the relative quantum yields of the forward and reverse reactions, not just a simple thermodynamic ranking. This is an oversimplified and slightly inaccurate explanation of PSS photochemistry.

Comment 9:

There are few typographical errors in the manuscript that need to be improved.

Author Response

Comment 1:

The manuscript (Page 6, lines 173-175) claims the two light-absorbing dipyrrole subunits of bilirubin "only differ structurally in the position of the methyl and vinyl substituents". This is a severe oversimplification and chemically inaccurate. The two chromophores are fundamentally non-equivalent due to the IXα structure (different methyl, vinyl, and propionic acid groups on C8,C12,C13,C17), which is the critical reason for the wavelength-dependent photochemistry discussed immediately afterward. 

Response 1: We appreciate the reviewer’s observation and agree that the underlying reasons for the differing formation rates of the various photoisomers (as discussed later in the manuscript) are indeed complex. However, the chemical difference between the two chromophores it is relatively straight forward, and we believe our description is chemically accurate in this context. The propionic acid subunits are symmetrically positioned within the bilirubin structure; therefore, the remaining difference lies in the placement of the vinyl and methyl substituents, as we noted and as the reviewer also acknowledges, these differ between two chromophores thus making them non-equivalent.  We have also added a reference that discusses this more in detail, see lines 174.

Comment 2: The manuscript (Page 9, lines 313-314) states that lumirubin (LR) is "irreversibly excreted via bile and predominantly via urine". While LR is water-soluble and cleared via the kidney, the primary physiological route for LR clearance in infants with normal liver function is via bile. The claim of "predominantly via urine" is generally considered inaccurate in the clinical context. 

Response 2: We have removed the word ‘predominantly’, and it now only reads “via bile and urine”, see line 318. 

Comment 3: The manuscript (Page 8, lines 272-274) includes δ-bilirubin (bilirubin covalently bound to albumin) in the list of species that react in the Direct Bilirubin (DBIL) fraction. This is conceptually confusing. δ-bilirubin does not react directly in the traditional diazo method (which defines DBIL) due to its strong covalent binding to albumin. It is measured in the "Direct" fraction only because its clearance pathway mimics conjugated bilirubin, but describing it as a reactive form in routine assays is chemically imprecise and misleading. 

Response3: We agree with the reviewer that the issue regarding reactivity of delta-bilirubin needs clarification. The delta billirubin has been found to be measured in both routine in vanadate assays, and diazo methods. The degree to which delta bilirubin contributes to “direct bilirubin” measured value varies with assay method, reagent composition, pH, buffer, reaction time. The lines 272-275 have been changed to clarify this issue

Comment 4: The manuscript (Page 8, line 279) states that in diazo methods, "only the configurational isomers" act as substrates. This is incorrect. Conjugated bilirubin (mono- and di-glucuronides) is the primary target that reacts in the direct fraction of the diazo method. While configurational isomers interfere with this measurement, they are not the only substrates, which fundamentally misrepresents the chemistry of the standard clinical assay

Response 4: We agree that the phrasing was not clear and could lead to misinterpretation. We have rephrased this sentence according to the reviewer suggestion as, “In addition to UCB and mono- and diconjugated forms of BR, both the configurational isomers and constitutional isomers (i.e., LR) can act as substrates in the vanadate method [54, 55], and the bilirubin oxidase method [56-58], whereas the configurational isomers but not LR can act as substrates in the diazo method [11, 22].” See lines 280-283.

Comment 5: The authors suggest that the unique structure of human albumin, which favors Z-LR formation, "has led some authors to suggest the possibility of an evolutionary step" to reduce the effects of neonatal hyperbilirubinemia. This is a highly speculative, teleological interpretation of natural variation and oversimplifies evolutionary processes, presenting an assumption as a widely accepted "suggestion" based on minor protein differences (Page 6, lines 207-210). 

Response 5: We thank the reviewer for this thoughtful comment and acknowledge that this point is indeed speculative. We have removed this sentence from the manuscript (see line 212).

Comment 6: The manuscript (Page 8, lines 290-291) claims the "inherent instability" of photoisomers means there is a "lack of available standards" for quantitative analysis. While purifying and maintaining pure, commercial standards for all isomers is challenging, this statement is too strong. Purified Z-LR standards are generated and used as critical reference points in the field, making the assertion of a total "lack" a misrepresentation of the practical state of the art. 

Response 6: We agree with the reviewer, and we did mean commercially available, we have added this to make it clearer, see line 292. 

Comment 7: The manuscript (Page 6, lines 194-196) claims that LR binding to HSA's subdomain IB "results in the increased binding capacity of albumin to UCB". The clearance of LR and ZE-BR frees up the secondary binding site (IB), which is beneficial for neuroprotection, but this process does not increase the total molar capacity or the affinity of the primary site (IIA) for unconjugated bilirubin (UCB). The claim overstates the observed neuroprotective effect. 

Response 7: We appreciate the reviewer comment, and have removed the sentence (see line 206) as it can be misunderstood to suggest increased affinity of UCB to the primary site (IIA) by merely binding LR in the subdomain IB. We shortened the sentence to emphasize the weak binding of ZE-BR to domain IB, and the reasonable functionality.

Comment 8: The manuscript (Page 6, lines 169-170) explains the equilibrium shift by stating that EE-BR is the "thermodynamically least favorable" configurational isomer. While EE-BR is the least stable (having no intramolecular hydrogen bonds), the composition of the PSS is governed by the relative quantum yields of the forward and reverse reactions, not just a simple thermodynamic ranking. This is an oversimplified and slightly inaccurate explanation of PSS photochemistry. 

Response 8: We thank the reviewer for this insightful comment. We agree that the discussion can indeed be framed in terms of Quantum yield. However, our intention was to highlight that quantum yield alone does not explain why a particular isomer forms to a lesser extent. A lower product yield in a photolytic reaction inherently implies a lower quantum yield, which reflects the probability of a given photochemical event occurring.

It is the instability of the EE form that shifts the relationship between the forward and reverse reaction more in favour of the reverse and subsequently formation of more stable isomers. Thus, our statement was not meant as a simplification, but rather to emphasize that product stability represents one of the key factors underlying the lower quantum yield observed for EE-BR.

Comment 9: There are few typographical errors in the manuscript that need to be improved

Response 9: We thank the Reviewer for noting the need for improving the English language. We have updated the manuscript accordingly.

Round 2

Reviewer 1 Report

Comments and Suggestions for Authors

The manuscript may be published in the journal in the present form.